# Analysis of Chemically Labile Glycation Adducts in Seed Proteins: Case Study of Methylglyoxal-Derived Hydroimidazolone 1 (MG-H1)

**DOI:** 10.3390/ijms20153659

**Published:** 2019-07-26

**Authors:** Kristina Antonova, Maria Vikhnina, Alena Soboleva, Tahir Mehmood, Marie-Louise Heymich, Tatiana Leonova, Mikhail Bankin, Elena Lukasheva, Sabrina Gensberger-Reigl, Sergei Medvedev, Galina Smolikova, Monika Pischetsrieder, Andrej Frolov

**Affiliations:** 1Department of Bioorganic Chemistry, Leibniz Institute of Plant Biochemistry, 06120 Halle, Germany; 2Department of Biochemistry, St. Petersburg State University, 199004 St. Petersburg, Russia; 3Department of Chemistry and Pharmacy, Food Chemistry, Friedrich-Alexander Universität Erlangen-Nürnberg (FAU), 91058 Erlangen, Germany; 4Department of Plant Physiology and Biochemistry, St. Petersburg State University, 199034 St. Petersburg, Russia

**Keywords:** Advanced glycation end products (AGEs), enzymatic hydrolysis, glycation, methylglyoxal-derived hydroimidazolone 1 (MG-H1), seeds, seed ageing, seed quality, sodium dodecyl sulfate (SDS)

## Abstract

Seeds represent the major source of food protein, impacting on both human nutrition and animal feeding. Therefore, seed quality needs to be appropriately addressed in the context of viability and food safety. Indeed, long-term and inappropriate storage of seeds might result in enhancement of protein glycation, which might affect their quality and longevity. Glycation of seed proteins can be probed by exhaustive acid hydrolysis and quantification of the glycation adduct *N^ɛ^*-(carboxymethyl)lysine (CML) by liquid chromatography-mass spectrometry (LC-MS). This approach, however, does not allow analysis of thermally and chemically labile glycation adducts, like glyoxal-, methylglyoxal- and 3-deoxyglucosone-derived hydroimidazolones. Although enzymatic hydrolysis might be a good solution in this context, it requires aqueous conditions, which cannot ensure reconstitution of seed protein isolates. Because of this, the complete profiles of seed advanced glycation end products (AGEs) are not characterized so far. Therefore, here we propose the approach, giving access to quantitative solubilization of seed proteins in presence of sodium dodecyl sulfate (SDS) and their quantitative enzymatic hydrolysis prior to removal of SDS by reversed phase solid phase extraction (RP-SPE). Using methylglyoxal-derived hydroimidazolone 1 (MG-H1) as a case example, we demonstrate the applicability of this method for reliable and sensitive LC-MS-based quantification of chemically labile AGEs and its compatibility with bioassays.

## 1. Introduction

Seeds represent the major source of food protein all over the world, and impact essentially on the daily human diet [1]. Therefore, seed quality, both in the sense of viability and food safety, needs to be secured. In this context, the conditions and duration of seed storage becomes an important factor, directly affecting seed quality [2]. It is known that prolonged or/and inappropriate storage of seeds results in dramatic enhancement of seed protein glycation, which is known to be an important marker of seed quality and longevity [3,4]. To some extent, this phenomenon can be simulated by a well-established model of accelerated ageing [5].

Glycation is a ubiquitous process of protein modification, accompanying interaction of amino and guanidino groups of polypeptides with carbonyl compounds (predominantly reducing sugars and α-dicarbonyls) [6], and yielding advanced glycation products (AGEs) [7]. These structurally diverse species are well-detectable in mammalian tissues [8,9] and are readily generated during thermal treatment of foods [10]. Accumulation of AGEs in mammalian organism results in intensive inter- and intra-molecular protein cross-linking [11], and triggers expression of molecular pro-inflammatory factors via interaction with multi-ligand immunoglobulin-like receptors (e.g., RAGEs—receptors to AGEs) [12]. Resulting sub-clinical inflammation accompanies ageing and related atherosclerotic changes in tissues [13], which ultimately contribute to the pathology of diabetes mellitus and neurodegenerative disorders, like Alzheimer and Parkinson diseases [14,15,16]. Last decade, protein glycation was reported in plants [17]. A deeper analysis of plant glycated proteome revealed a strong prevalence of AGE formation over early glycation [18]. Moreover, glycation of specific proteins was enhanced not only in presence of environmental stressors, like drought [19], but also accompanied ageing in leaves [20], legume nodules [21] and seeds [22]. 

Methylglyoxal (MGO)-derived hydroimidazolones are the products of non-enzymatic reaction of this α-dicarbonyl with arginyl residues of proteins, and are represented with three isomers, namely—*N*^δ^-(5-methyl-4-oxo-5-hydroimidazo-linone-2-yl)-l-ornithine (MG-H1, the major adduct) [23], 2-amino-5-(2-amino-5-hydro-5-methyl-4-imidazolon-1-yl)pentanoic acid (MG-H2) and 2-amino-5-(2-amino-4-hydro-4-methyl-5-imidazolon-1-yl)pentanoic acid (MG-H3) [24]. According to the formation pathway, proposed by Glomb and co-workers [25] and confirmed at the peptide level [26], reaction of arginyl residues with methylglyoxal yields an intermediate—methylglyoxal-derived dihydroxyimidazolidine (Figure 1). Its dehydration results in formation of MG-H3, which can be reversibly hydrolyzed to *N^δ^*-(carboxymethyl)arginine under alkaline conditions [27]. Further re-condensation and formation of the hydroimidazolone ring yields the major product MG-H1, whereas involvement of the second MGO molecule in the reaction with dihydroxyimidazolidine results in formation of *N*^δ^-(5-hydroxy-4,6-dimethylpyrimidine-2-yl)-l-ornithine (argpyrimidine) [28] and *N*^δ^-(4-carboxy-4,6-dimethyl-5,6-dihydroxy-1,4,5,6-tetrahydropyrimidine-2-yl)-l-ornithine(tetra-hy-dropyrimidine) [29].

Quantification of free MG-H isomers in biological samples typically relies on chromatographic techniques with fluorescence [30] or tandem mass spectrometry (MS/MS)-based detection [31]. To assess the contents of protein-bound adducts, exhaustive hydrolysis of corresponding polypeptides needs to be done [9]. Thereby, standard isotope dilution and standard addition represent the “gold standard” in quantitative analysis of AGEs in biological matrices [32]. In the most easy and straightforward way, quantitative hydrolytic degradation of any intra- and extra-cellular protein can be achieved by acid hydrolysis in presence of 6N HCl at 100–110 °C (18–24 h) [33,34]. This method is well-applicable to seeds, and allows complete degradation of total protein isolates, which are typically only partly soluble in aqueous solutions and cannot be adequately addressed by enzymatic techniques. However, as up to 90% hydroimidazolones degrade under conditions of acid hydrolysis [24], analysis of these products by multi-step enzymatic degradation procedures represents the only option to address AGE contents in biological samples. Whereas for some insoluble proteins, like collagen, this can be achieved [35], for complex multi-protein mixtures, like preparations of total seed protein, it could not be done so far. Therefore, here we propose a modified enzyme-based protocol, giving access to quantitative hydrolysis of total seed protein, and compatible both with LC-MS-based absolute quantification techniques and biological assays.

## 2. Results

### 2.1. Protein Isolation and Enzymatic Hydrolysis

As pea seeds represent a protein-rich matrix, the protein extraction method was optimized in respect to the tissue amounts, taken for the phenol extraction. The maximal protein recovery (91.6 ± 22.5 mg protein/g fresh weight) could be achieved with the sample amount of 50 mg (Appendix A). Therefore, pea proteins were isolated from approximately 50 mg of frozen embryos (*n* =3). The oilseed rape seeds (*n* = 5) were isolated in two portions, which were combined afterwards.

As expected, the protein isolates could be only partly reconstituted in aqueous buffers (Tris-HCl buffer or phosphate buffered saline, PBS). Therefore, to achieve quantitative protein hydrolysis, we decided for reconstitution of the isolates in presence of a detergent (10–20% (*v*/*v*) Triton X-100 or 10% (*w*/*v*) sodium dodecyl sulfate, SDS), which was supposed to be diluted prior to hydrolysis and removed afterwards. While application of Triton X-100 did not result in complete reconstitution of protein pellets, it was the case with the SDS solution. Determination of protein contents in aq. 10% (*w*/*v*) SDS revealed the extraction yields in the range of 39.6–66.6 and 19.4–28.4 mg/g fresh weight for pea and oilseed rape seeds, respectively (Appendix A). The assay precision was determined by SDS-PAGE loading 5 µg of protein on each lane: the overall lane densities were 3.1 × 10^4^ ± 6.0 × 10^3^ arbitrary units (AU, RSD = 19.3%) and 2.9 × 10^4^ ± 9.2 × 10^2^ AU (RSD = 3.2%) for pea and oilseed rape, respectively.

After a 20-fold dilution with phosphate buffered saline, the resulted concentration of SDS (0.5% *w*/*v*) was compatible with the activities of pronase E, proteinase K and carboxypeptidase Y. Therefore, the protocol, established in the Glomb’s group [9], could be successfully employed here. SDS-PAGE showed no protein signals already after the first incubation with pronase E (Appendix A). The applicability of the method was further confirmed with heavily glycated bovine serum albumin (BSA, Appendix A). Syringe infusion analysis of the aliquots sampled after the completion of the hydrolysis (but before SPE), revealed no multi-charged signals, potentially corresponding to non-digested peptides (Appendix A). Thus, the hydrolysis of seed protein could be considered to be exhaustive and quantitative.

### 2.2. Removal of the Detergent by Solid Phase Extraction

For the removal of the detergent we considered solid phase extraction (SPE) on three CHROMABOND materials–reversed phase (C18), weak and strong anion exchangers (HR-XAW and HR-XA, respectively). For this, mixtures of three basic amino acids, known as precursors of AGEs (lysine, arginine and histidine) were loaded on all three cartridges according to the instructions of the producer (Protocols S1–3). After derivatization with *N*^2^-(5-fluoro-2,4-dinitrophenyl)-l-valine amide (L-FDVA) according a well-established protocol [36], amino acid derivatives were analyzed by RP-UHPLC-ESI-LIT-Orbitrap-MS. Among the three tested SPE protocols, the strong anion exchanger showed poor recovery for lysine and histidine (16% and 45% in combined flow-through and wash fractions, respectively, Appendix A), whereas the weak anion exchanger showed good recoveries for all analytes, but a strong signal enhancement effect for lysine was observed (Appendix A). FIA-HR-MS analysis indicated quantitative retention of SDS by both phases. The reverse phase cartridges showed good recovery for histidine and arginine, whereas for lysine it was slightly compromised (57%, Appendix A). Thereby, to ensure quantitative elution of all amino acids, two wash steps were used in this system—with 25 mmol/L ammonium acetate and with 100 mmol/L ammonia. FIA-HR-MS analysis showed that SDS was quantitatively retained on the cartridge in absence of acetonitrile in the eluent. However, only 10% (*v*/*v*) acetonitrile in eluent mixture resulted in partial elution of SDS from the cartridge and contamination of the amino acid sample. Remarkably, when 200 µg of hydrolyzates, obtained from glycated BSA were loaded on C18 cartridges with and without spiking with 0.5% (*w*/*v*) SDS (*n* = 4), no effect of SDS on signal intensities in subsequent LC-MS analyses was observed (Appendix A).At the next step we considered the performance of C18 cartridges with and without endcapping. For this, 200 µg of glycated BSA in 1 mL of PBS were loaded on CROMABOND C18 cartridges with and without endcapping either with or without pre-spiking with SDS solution to get final concentration of 0.5% (*w*/*v*). The result indicated slightly better performance of endcapped cartridges for both MG-H1 and arginine (Appendix A, respectively), whereas for a hydrophobic amino acid phenylalanine an enhancement effect was observed (Appendix A). The recoveries of CML (taken as a reference AGE) and lysine were also good, although signal enhancement for lysine was observed (Appendix A). Thus, the final SPE protocol relied on C18 endcapped cartridges and a two-step elution procedure, comprising 25 mmol/L ammonium acetate and with 100 mmol/L ammonia (6 mL each).

### 2.3. RP-UHPLC-ESI-LIT-Orbitrap-MS Analysis

Having the optimized sample preparation procedure in hand, we established a method for absolute quantification of MG-H1 in biological matrices, as a case proof of the concept experiment. As stable isotope dilution technique is a gold standard in analysis of glycated adducts [37], we decided on this approach here. For this, first, the amounts of MG-H1 were roughly estimated by external calibration with the authentic standard. After adjustment of the sample loading conditions (in respect to the weight of hydrolyzed protein), the amounts of MG-H1 in 100 µL of injected hydrolyzates of glycated BSA and pea seed protein extracts were found to be 71 and 1 pmol, respectively. Therefore, we adjusted the amounts of the spiked internal standard (MG-H1-d3) to ensure the intensity ratio of analyte and internal standard within two orders of magnitude around 1.0. Thus, quantitative analysis could be reliable, when 10–50 pmol of internal standard were loaded on the column.

The internal standard suited well to the proposed quantification strategy. Indeed, on one hand it co-eluted with the analyte, on another, it could be clearly distinguished from the analyte by the isotopic pattern (Figure 2). Therefore, extracted ion chromatograms (XICs) could be independently calculated from both compounds, and intensity ratio could be determined in a simple and straightforward way. Tandem mass spectrometric (LIT-Orbitrap-MS/MS) analysis delivered well-interpretable fragmentation patterns (Figure 3A), which were identical for the analyte and commercially available authentic MG-H1 standard (Figure 3B). The MS/MS spectrum of the internal standard (MG-H1-d3) also reproduced the spectrum of analyte with consideration of the deuterium substitution (Figure 3C).

### 2.4. Standardization and Validation of the Quantification Method

The instrument LODs and LOQs of MG-H1 (determined as MG-H1-d3 spiked to BSA hydrolysate) were 2.5 and 25 fmol, respectively with linear dynamic range (LDR) of 4 × 10^4^ (Table 1). It was comparable with these parameters obtained for reference amino acids—arginine, lysine, alanine and phenylalanine (5–25 fmol, 25–100 fmol, and 0.1–1.0 × 10^3^, respectively, Appendix A). Thereby, the coefficient of determination (R^2^) was not less than 0.988. The method showed an excellent intra- and inter-day precision for both derivatization procedure and LC-MS analysis itself, as well as for the overall sample preparation procedure, comprising the whole enzymatic hydrolysis and SPE. Indeed, the first two values were about 1% (Table 2 and Table 3), whereas the overall precision hydrolysis/SPE precision was below 8.0% (Table 4). Importantly, L-FDVA derivatives of MG-H1 showed good stability under the conditions of autosampler (8 °C): the alterations of the analyte recoveries were within 4%, when repetitive injections from the same vial were performed during three consecutive days (*n* = 4/day).

For analysis of recovery, MG-H1-d3 was spiked to isolated pea protein in aq. 10 % (*w*/*v*) SDS prior to dilution with PBS and enzymatic hydrolysis. To address the interference of the target analyte with the components of protein hydrolyzates, the recovered amounts of MG-H1-d3 were compared to those, observed when the same aliquots of MG-H1-d3 were spiked to aq. 10 % (*w*/*v*) SDS, subjected to the identical treatment. To address the effect of incubations with the enzymes on glycation adduct stability, the recovered amounts of MG-H1-d3 were compared to the standard solution of MG-H1-d3, dissolved in PBS at the same concentration. The analysis revealed relatively low contribution of other hydrolysis products (i.e., highly abundant proteinogenic amino acids): a slight enhancement of detection was observed in comparison to blank hydrolysis samples (recovery 109.4%, Appendix A). In contrast, the prolonged hydrolysis procedure itself resulted in a relatively high degree of the target analyte suppression or degradation—only 18% of MG-H1-d3 could be recovered, although this result was reliably reproducible (Appendix A).

For analysis of the possible matrix effects as a potential reason for the observed recovery loss, the approach of Böttcheret al. [38] was applied. For this, a L-FDVA derivative of the MG-H1-d3 standard (0.2 µmol/L) was infused with a syringe pump at 3 µL/min in the effluent of RP-HPLC column after injection of blank (20% *v*/*v* aq. acetonitrile in 0.1% *v*/*v* formic acid, Appendix A) or derivatized pea protein hydrolyzate (13.6 µg, Appendix A). Although no matrix effects could be detected in blank (Appendix A), approximately complete suppression of the MG-H1-d3-L-FDVA signal was observed at t_R_ 8.8–9.2 min and 11.5–12.0 min when the standard was infused in the column effluent obtained with pea protein hydrolyzate (Appendix A). Thus, the section of the chromatogram, corresponding to t_R_ of MG-H1 and its labeled counterpart (t_R_ = 12.3 min) was only minimally affected by ion suppression, i.e., the observed loss of recovery upon the hydrolysis procedure is mostly related to the compromised stability of the hydroimidazolone glycation adduct under the prolonged hydrolysis conditions.

### 2.5. Quantification of MG-H1 in Seed Protein Hydrolysates by Stable Isotope Dilution

To address the applicability of the developed quantification approach to determination of MG-H1 contents in seed proteins, we analyzed protein hydrolysates obtained from pea and oilseed rape seeds. The analysis revealed more than eight-fold higher MG-H1 levels, present in pea seeds in comparison to those determined in the oilseed rape seeds (1.31 ± 0.022 vs. 0.17 ± 0.01 nmol/mg protein, Figure 4). Unexpectedly, accelerated ageing did not reveal any increase of MG-H1 seed contents (Figure 4A). In contrast, a non-significant tendency for down-regulation of this AGE was observed. On the other hand, natural ageing during nine years resulted in a slight and non-significant up-regulation of MG-H1 in the oilseed rape seeds (Figure 4B).

### 2.6. Compatibility of the Hydrolysis Protocol with Cell Assays

To address the compatibility of the optimized hydrolysis protocol with biological assays, we characterized toxicity of the obtained hydrolysates for cultured neuroblastoma SH-SY5Y cells. The main aim of this section was to verify, that the degree of SDS removal with our SPE procedure is sufficient for cell assays, i.e., that residual SDS does not exert cytotoxicity, or such effects are negligible. For this, 2 mg of lyophilized glycated BSA and protein, isolated from pea seeds, not subjected to accelerated ageing, were reconstituted in 50 µL of 10% (*w*/*v*) aq. SDS and hydrolyzed as described in the Materials and Methods section. Pre-cleaned hydrolysates were freeze-dried, reconstituted in medium and different amounts of isolates were applied to cultured cells. For glycated BSA, the MTT assay did not reveal any statistically significant difference from untreated controls up to the concentration of 0.4 mg/mL (Figure 5A), whereas for pea protein a minimal reduction of cell viability (17.1%) was observed with 0.3 mg/mL (*t*-test: *p* = 0.021), with no effect observed at lower concentrations (Figure 5B).

## 3. Discussion

Protein glycation impacts essentially on seed quality and longevity. Therefore, correct and comprehensive characterization of the seed protein glycation profiles is absolutely mandatory for understanding the changes in seed metabolism, accompanying stress-related alterations during maturation and storage of seeds. Although proteomics might deliver important functional information, adequate absolute quantification of a broad panel of glycated amino acid adducts is necessary to understand the underlying chemistry of Maillard reaction. However, the related currently existing amino acid analysis workflows suffer from two bottlenecks: on one hand, the majority of glycation adducts are acid- and/or thermally labile, on the other hand—implementation of enzymatic hydrolysis under physiological conditions is restricted by low solubility of seed protein isolates in aqueous buffers.

### 3.1. Solubilization and Enzymatic Hydrolysis

It is well-known, that glycation, i.e., Maillard reaction of proteins, can contribute to seed ageing via protein modification and suppression of cellular antioxidant defense [39]. Although glycation of seed proteins can be addressed by spectrofluorometry [40] and ELISA [22], the state-of-the-art level of analytical techniques assumes LC-MS- or LC-MS/MS-based quantification of multiple specific glycation adducts [37].Some AGEs, like CML or pentosidine, can be readily analyzed in different matrices due to efficient solubilization during acid hydrolysis [9]. For some potentially insoluble proteins, like collagen, solubilization can be achieved simultaneously with enzymatic degradation by using such enzymes as collagenase and pepsin [7,41]. However, as no ambient protocol for solubilization of seed proteins is reported so far, analysis of chemically labile seed hydroimidazolone AGEs and related structures is still not reported, although it is known, that glycation reduces susceptibility of pear seed protein to enzymatic hydrolysis [42].

The main reason for this is low solubility of seed protein isolates. Indeed, legume seed proteome is strongly dominated by several highly abundant protein families—vicilins, convicilins, 11S globulins (legumins), and 2S albumins (PA1 and PA2) [43], whereas the seed proteome of mature *Brassica napus* seeds at least for 20% is represented by napin [44]. Despite potential solubility of the major seed proteins in water, their isolation is usually accompanied with aggregation with minor hydrophobic seed polypeptides and non-protein constituents that result in low solubility of such preparations. In standard proteomic practice, such total protein isolates can be reconstituted in high concentrations of chaotropic reagents (urea, thiourea) of in presence of detergents—conventional (SDS, triton X100) or degradable (e.g., RapiGest or Anionic Acid-Labile Surfactant, AALS) and effectively digested, followed with removal of solubilization agents [3].

In line with these considerations, we decided to transfer the described approach to the exhaustive enzymatic hydrolysis. Accordingly, we achieved complete reconstitution of protein in a small volume of 10% (*w*/*v*) SDS diluted it to accomplish hydrolysis and removed SDS by SPE afterwards. This procedure turned out to be ideally compatible to the enzymatic digestion protocol of the Glomb’s group [9]. Indeed, all three enzymes—pronase E, proteinase K and carboxypeptidase Y, used by these authors, preserved their activity in presence of 0.5% SDS [45,46,47,48,49]. Thus, reconstitution of protein pellets in 50 µL of 10% (*w*/*v*) SDS with subsequent 20-fold dilution with PBS directly brought us to the starting point of this well-established and reliable protocol. Remarkably, ESI-MS analysis of the obtained hydrolysates revealed complete absence of peptides in the mixtures (Appendix A) that indicated completeness of digestion and no necessity in post-hydrolysis filtration step, as it is often done [24].

### 3.2. Removal of the Detergent from Hydrolysates

Selective removal of SDS from the protein hydrolysates was based on the difference in properties of analytes and SDS: even under highly alkali aqueous conditions SDS was completely retained on the reversed phase, whereas amino acids were quantitatively eluted. Thereby, we used two elution steps. The first one, accomplished with the 25 mmol/L ammonium acetate solution, targeted basic amino acids and adducts, poorly retained on the reversed phase (e.g., hydroimidazolones and, in particular; target compound MG-H1). The second eluent was a base, providing quantitative deprotonation of carboxyl functions and quantitative elution of hydrophobic amino acids, like phenylalanine and tryptophan, which typically retain well on the reversed phase under acidic conditions. This scheme might allow avoiding on-cartridge conversion of alkali-labile imidazolones MG-H3 and Glarg in *N^δ^*-(carboxyethyl)-arginine (CEA) and *N^δ^*-(carboxymethyl)-arginine (CMA), respectively [27]. The immediate freezing of the eluates after the second elution was, in this context, desired.

### 3.3. Quantitative Analysis

The principle changes, done in the original hydrolysis protocol, ultimately required verification of compatibility with existing quantitative techniques and comprehensive validation of the overall procedure. As the main scope of this study was extending the applicability of existing analytical techniques to new objects and matrices, but not development of new LC-MS or LC-MS/MS protocols, we used here pre-column derivatization with *N*^2^-(5-fluoro-2,4-dinitrophenyl)-l-valine amide (L-FDVA) with subsequent LC-MS analysis—the method established in our group more than a decade ago [36].Here we just adjusted gradient to the UHPLC technique and optimized the MS method for the LIT-Orbitrap hybrid.

Surprisingly, our method turned out to be rather sensitive. Thus, despite that a relatively old Orbitrap Elite instrument, operated in the full scan mode, was used here, the limit of detection for MG-H1 (determined with MG-H-d3, spiked to the hydrolysate of glycated BSA, prepared according to our procedure, Table 1) was even lower, than with a triple quadrupole (QqQ)-based multiple reaction monitoring (MRM) method of Hashimoto et al, relying on derivatization with 2,4,6-trinitrobenzene sulfonate [50]. Thereby, an excellent linear dynamic range (LDR), covering four orders of magnitude, could be achieved for this glycation adduct. Most probably, it can be attributed to decrease of matrix-related suppression effects [51] for cationic derivatives after the SPE procedure, of even signal enhancement, as could be seen in model experiments with amino acid mixtures (Appendix A) and glycated BSA (Appendix A). The enhancement effect can be, at least partly, underlied by formation of ammonia adducts in eluates and, therefore, lower losses of analytes by adsorption on the walls of polypropylene tubes. Of course, the method sensitivities can be essentially increased by switch to well-established MRM-based workflows [52,53]. Establishing a new MS/MS-based protocol was, however, behind the scope of this study. Not less importantly, one needs to take into account that, due to high contents of ammonium acetate in samples, our workflow can potentially affect analyte retention in ion pair chromatography. Indeed, due to a relatively low flow rate, a local drop of the ion pair reagent concentration in the eluent can occur, and retention time shifts can be detected (personal unpublished observation). Thereby, higher injection volumes (as, for example, 100 µL used here) would favor these effects. Indeed, acetate might form a strong ion pair with trifluoroacetic or heptafluorobutyric acids and change retention times and peak symmetry. Clearing this issue must also be also the matter of future studies. As reduction of injection volumes seems to be not possible in terms of sensitivity, an additional SPE step might be considered.

The most remarkable feature of our procedure was its high intra- and inter-day precision. Indeed, the overall relative standard deviation (RSD%) of hydrolysis and SPE did not exceed 8%, whereas the precision of the derivatization procedure and the measurement itself was within 2%, that is relatively low for LC-MS-based quantification [52]. Hence, the digestion/pre-cleaning method block can be incorporated in any MS-based protocol, independently from employed derivatization and/or chromatographic system. Moreover, it can be incorporated in functional physiological assays of Maillard reaction products, and provide thereby direct access to structure-response relationships.

## 4. Materials and Methods

### 4.1. Reagents and Plant Material

Unless stated otherwise, materials were obtained from the following manufacturers. Carl Roth GmbH & Co. (Karlsruhe, Germany): ammonia solution (25%); ethanol (≥99.8%), sodium dodecyl sulfate (SDS) (>99%), sodium chloride (p.a.), sodium phosphate dibasic dehydrate (p.a.); PanReac AppliChem (Darmstadt, Germany): acrylamide (2K Standard Grade), glycerol (ACS grade); AMRESCO LLC (Fountain Parkway Solon, OH, USA): ammonium persulfate (ACS grade), glycine (biotechnology grade), *N*,*N*′-methylene-bis-acrylamide (ultra pure grade), *tris*(hydroxymethyl)aminomethane (tris, ultra pure grade), urea (ultra pure grade); Component-Reactiv (Moscow, Russia): phosphoric acid (p.a.); Iris Biotech GmbH (Marktredwitz, Germany): methylglyoxal-derived hydroimidazolone 1 (MG-H1, p.a.) and MG-H1-d3 (p.a.); Merck KGaA (Darmstadt, Germany): potassium chloride (p.a.), sodium phosphate monobasic monohydrate (p.a.); Reachem (Moscow, Russia): hydrochloric acid (p.a.), isopropanol (reagent grade); SERVA Electrophoresis GmbH (Heidelberg, Germany): Coomassie Brilliant Blue G-250, 2-mercaptoethanol (research grade); Thermo Fisher Scientific (Waltham, USA): Pierce™ Unstained Protein Molecular Weight Marker #26610 (14.4–116.0 kDa), PageRuller™ Plus Prestained Protein Ladder #26620 (10–250 kDa); Vekton (St. Petersburg, Russia): acetonitrile (HPLC grade), conc. HCl (puriss). All other chemicals were purchased from Sigma-Aldrich Chemie GmbH (Taufkirchen, Germany). Water was purified in house on a water conditioning and purification system Millipore Milli-Q Gradient A10 system (resistance 18 mΩ/cm, Merck Millipore, Darmstadt, Germany).

Pea seeds of the cultivar Millennium were obtained from the Research and Practical Center of National Academy of Science of the Republic of Belarus for Arable Farming (Zhodino, Belarus, harvested in the year 2015 and stored at 18 °C). Oilseed rape seeds of the cultivar Oredezh-2 (K-4917) from the State research enterprise Leningrad Research institute for Applied Agricultural Science “Belogorka” Russian Academy of Agricultural Science were provided by the Federal Research Center “The N.I. Vavilov All-Russian Institute of Plant Genetic Resources” (VIR). Rape seeds were harvested in the year 2008 and 2015 and stored at 18 °C.

### 4.2. Glycation of Bovine Serum Albumin (BSA)

Glycation of BSA was accomplished by a well-established procedure [54], later modified by Greifenhagen et al [55] and slightly changed here. In detail, 20 mg of BSA in 1 mL of 100 mmol/L sodium phosphate buffer (pH 7.4), containing 18 μmol/L FeSO_4_ and 250 mmol/L d-glucose, was filtrated with a 0.22 µm filter and incubated at 55 °C during 7 days under continuous shaking (450 rpm) in safe-lock 1.5 mL polypropylene tubes. After the completion of the incubations, the buffer was changed to phosphate buffered saline (PBS) by ultrafiltration in Vivaspin filter devices equipped with polyethersulfone (PES) membrane with 3000 MW cut-off (Sartorius, Göttingen, Germany). Afterwards, protein concentration was determined by 2D-Quant kit (GE Healthcare, Taufkirchen, Germany) as described by Matamoros and co-workers [21].

### 4.3. Ageing of Pea and Oilseed Rape Seeds

To establish the model of natural ageing, rape seeds were stored for nine years at 18 °C in the dark. To establish the model of accelerated ageing, pea seeds were incubated in a desiccator at 45°C above a saturated KCl solution (86% relative humidity). It resulted in increase of the seed water content from 9% to 18% (*w*/*w*). The seeds were removed from the desiccator on the fifth day, dried to the initial water content of 9% (*w*/*w*), and stored at 4 °C in closed containers.

### 4.4. Protein Isolation

Pea and oil seed rape seeds (10 and 35 per biological replicate, for pea and oilseed rape, respectively, *n* = 3) were frozen in liquid nitrogen and ground in a Mixer Mill MM 400 ball mill with a Ø 20 mm stainless steel ball (Retsch, Haan, Germany) at a vibration frequency of 30 Hz for 2 × 1 min. The obtained frozen powder (approximately 50 mg per replicate) was transferred to 2 mL safe-lock polypropylene tubes and stored at −80 °C prior to protein isolation, which relied on the phenol extraction method, as described by Frolov and co-workers [56] with some modifications. Briefly, the tubes with seed material were placed on ice, and 650 µL of cold (4 °C) phenol extraction buffer (0.7 mol/L sucrose, 0.1 mol/L KCl, 5 mmol/L ethylenediaminetetraacetic acid (EDTA), 2% (*v*/*v*) mercaptoethanol and 1 mmol/L phenylmethylsulfonyl fluoride (PMSF) in 0.5 mol/L tris-HCl buffer, pH 7.5) were added. The suspensions were vortexed for 30 s, and 650 µL of cold phenol (4 °C) saturated with 0.5 mol/L tris-HCl buffer (pH 7.5) were added. Afterwards, the samples were extracted for 30 min at 900 rpm (4 °C) and centrifuged (5000× *g*, 30 min, 4 °C). The phenolic (upper) phases were transferred to new 1.5 mL polypropylene tubes and washed two times with equal volumes of the phenol extraction buffer (with vortexing 30 s, shaking for 30 min at 900 rpm at 4 °C and centrifugation at 5000 g for 15 min at 4 °C after each buffer addition). Finally, the supernatants were collected in 1.5 mL polypropylene safe-lock tubes, and proteins were precipitated by a five-fold volume of cold 0.1 mol/L ammonium acetate in methanol, followed with storage at −20 °C overnight. After this, the proteins were pelleted by centrifugation (10 min, 5000× *g*, 4 °C). The pellets were washed twice by re-suspending in two volumes of methanol (compared to the phenol phase volume), and twice—in the same volume of acetone (both at 4 °C), followed with centrifugation (5000× *g*, 10 min, 4 °C). The final pellets were dried under air in a hood for 1 h, re-dissolved in 100 µL of 10% (*w*/*v*) SDS, and protein contents were determined by 2-D Quant Kit.

### 4.5. SDS-PAGE

Polyacrylamide gel electrophoresis in sodium dodecyl sulfate (SDS-PAGE) was performed as described by Greifenhagen et al. [57] with minor modifications. In detail, separations were performed with a 12% resolving and a 6% stacking gel (T = 12%, C = 2.65%).The aliquots of protein or hydrolysate samples (10 µg), were freeze-dried under reduced pressure and reconstituted in 20 µL of sample buffer, containing 0.05% (*w*/*v*) bromophenol blue, 20% (*v*/*v*) glycerol, 2% (*w*/*v*) SDS, 5% (*v*/*v*) β-mercaptoethanol in 62.5 mmol/L Tris-HCl (pH 6.8). On each lane, 10 µL of dissolved sample (5 µg each) were loaded. Aliquots of enzymatic hydrolysates, corresponding to 5 µg, were completely dried under reduced pressure, reconstituted in 10 µL of the same buffer and completely loaded on each lane. The molecular weights of individual proteins were assigned by a molecular weight standard mix, run on the same gel. After completion of separation (45 min at 200 V), gels were stained with Coomassie Brilliant Blue G-250 for 1 h. Average densities across individual lanes (expressed in arbitrary units) were determined by ChemiDoc XRS imaging system controlled by Quantity One^®^ 1-D analysis software (Bio-Rad Laboratories Ltd., Moscow, Russia). For calculation of relative standard deviations (RSDs), the densities of individual lines were normalized to the gel average value.

### 4.6. Exhaustive Enzymatic Hydrolysis

Enzymatic hydrolysis was performed with 0.3–3.7 mg of BSA, 1 mg of oilseed rape and 0.3 mg of pea protein. The appropriate volumes of protein solutions in 10% (*w*/*v*) SDS (not exceeding 50 µL) were transferred in a 2 mL safe-lock polypropylene tube and adjusted to 1 mL with phosphate buffered saline (PBS, 137 mmol/L NaCl, 2.7 mmol/L KCl, 10 mmol/L Na_2_HPO_4_, 1.8 mmol/L KH_2_PO_4_, pH 7.4). The resulted mixtures were supplemented with internal standard (560 pmol of MG-H-d3), 3 µL of 1 mol/L CaCl_2_ solution and a small crystal of thymol. Then, the following enzymes were added sequentially: 1.2 units of pronase E (twice), 0.195 units of proteinase K and 0.05 units of carboxypeptidase Y. Thereby, all incubations were performed for 24 h at 37 °C (incubation with carboxypeptidase Y was performed at 25 °C) under continuous shaking at 450 rpm in the dark.

### 4.7. Solid Phase Extraction (SPE)

Solid phase extraction (SPE) was done on reversed phase using CHROMABOND ec (end-capped) C18 cartridges and vacuum manifold (Macherey Nagel, Düren, Germany, operated under the pressure of 850 mbar. The cartridges were pre-conditioned with 6 mL of methanol, equilibrated with 6 mL of water, before the hydrolysates (1 mL each) were applied. The fraction, containing amino acids was sequentially eluted with 6 mL of 25 mmol/L aq. ammonium acetate and 6 mL of 100 mmol/L aq. ammonia. The flow-through and both eluate fractions were saved in one 15-mL polypropylene tube and freeze-dried. The residues were sequentially reconstituted in two 0.5-mL portions of 20% (*v*/*v*) aq. acetonitrile, transferred to a 1.5-mL polypropylene tube, freeze-dried and stored at −20 °C before analysis.

### 4.8. Derivatization

Prior to analysis, hydrolysates were reconstituted in 30 µL of 20% (*v*/*v*) acetonitrile and derivatized with *N*^2^-(5-fluoro-2,4-dinitrophenyl)-l-valine amide (L-FDVA) as described by Ehrlich and co-workers [36] with some modifications. In detail, 20 µL of hydrolysate aliquots were supplemented with 7 µL of water and pH was adjusted to 8.0 with 1 mol/L NaHCO_3_ using indicator paper (typically 4–20 µL of NaHCO_3_ were required). Afterwards, 32 µL of 36.7 mmol/L *N*^2^-(5-fluoro-2,4-dinitrophenyl)-l-valine amide (L-FDVA) acetone solution were added, and reactions were performed during 90 min at 40 °C under continuous shaking (350 rpm), before the reactions were stopped by addition of 25 µL of 1 mol/L HCl. After the change of solution color to yellow, 50 µL of acetonitrile and 96 µL of water were added, and the samples were intensively vortexed. Finally, 500 µL 0.1 % (*v*/*v*) formic acid were added, the samples were centrifuged (15,000× *g*, 5 min, room temperature), and the supernatants were transferred to the inserts of the vials for HPLC. 

### 4.9. RP-UHPLC-ESI-LIT-Orbitrap-MS Analysis

For analysis of the L-FDVA derivatives in hydrolyzates, 100 µL of sample were loaded on a reversed phase Hypersil GOLD aQ column (100 × 1 mm, 1.9 µm particle size), installed on a Dionex Ultimate 3000 UHPLC System (Thermo fisher Scientific, Bremen, Germany). The separations were performed at the flow rate of 150 µL/min, at 40 °C in a linear gradient mode, with eluents A and B being water and acetonitrile, both containing 0.1% (*v*/*v*) formic acid. After a two-minute isocratic step at 5% eluent B, amino acid derivatives were separated in the sequential gradients to 20%, 33% and 42% eluent B in 1, 7, and 4 min, respectively. After a second isocratic segment (4 min at 42% eluent B) a further gradient to 70% eluent B was run in 3 min. The column effluents were transferred on-line into a hybrid LTQ-Orbitrap Elite mass spectrometer (Thermo Fisher Scientific, Darmstadt, Germany), equipped with a heated electrospray ionization (HESI) source and operated in the positive ion mode, under the settings, listed in Appendix A. Annotation of analytes relied on *m*/*z*, t*_R_* and MS/MS data. Quantitative analysis relied on the stable isotope dilution strategy and integration of corresponding extracted ion chromatograms (XICs) at specific t_R_. Peak integration was performed in LCquan™ 2.8 software (Thermo Fisher Scientific, Berlin, Germany).

### 4.10. Method Validation

In terms of the method validation, limits of detection and quantification (LOD and LOQ, respectively), linear dynamic range (LDR), accuracy, stability, as well as intra- and inter-day precision for hydrolysis, derivatization, and chromatographic analysis, were determined. The analysis of hydrolysis precision relied on spiking of 500 µg (*n* = 12, 4 replicates per day) glycated BSA with the internal standard (MG-H1-d3, 0.5 µmol/L) prior to hydrolysis. The other precision tests and stability assay were performed with 300 µg aliquots of BSA (*n* = 4 per day, totally 12 assays/test) and 0.25 µmol/L spiked standard. For each precision test, an appropriate pooled sample was prepared, aliquoted and frozen. Four aliquots were thawed and analyzed on each of three consecutive days, and intra-/inter-day precision was determined. For the assessment of analyte stability, the same samples were analyzed for three constitutive days.

### 4.11. Cell Culture

Human mock-transfected neuroblastoma SH-SY5Y cells were cultured in a high glucose (4.5 g/L) Dulbecco’s modified Eagle’s medium (DMEM) with glutamax supplied with 10% heat-inactivated fetal calf serum, 1% penicillin and streptomycin, 1% minimal essential media vitamin, 1% nonessential amino acid, 1% sodium pyruvate, and 0.66% hygromycin B at 37 °C, 5% CO_2_, 95% air. Cells were passaged by trypsinization. The procedure was conducted by rinsing the cells with PBS at 37 °C before detaching cells by 3 mL of trypsin/EDTA solution (0.25%/0.2%) at 37 °C for 2 min. Cell detachment was stopped by adding of 10 mL of full medium. After that cells were collected by centrifugation (1500 rpm, 5 min, 25 °C) and re-suspended in fresh medium. The cell number was determined using an automatic cell counter. For 3-(4,5-dimethylthiazol-2-yl)-2,5-diphenyltetrazolium bromide (MTT) assay, cell confluence of ≥80% and passages from 4 to 10 were used.

### 4.12. Analysis of Cell Viability by MTT Assay

The viability of human neuroblastoma cells SH-SY5Y was determined by a quantitative colorimetric assay with 3-(4,5-dimethyl thiazol-2-yl)-2,5-diphenyltetrazolium bromide (MTT) according to the method of Mosmann [58] with modifications. The cells were seeded in 96-well plates at a density of 50,000 cells /100 µL per well and were allowed to grow for 48 h before treatment. Then the medium was changed, and the cells were stimulated with sterile filtrated protein hydrolysates for 24 h at 37 °C. Concentration of tested substances ranged from 0.05 to 0.4 mg/mL for the glycated hydrolyzed BSA and from 0.075 to 0.3 mg/mL for the hydrolyzed pea seed protein, dissolved in culture medium. Control cells were incubated with medium without further supplements. The cells treated with 10% (*v*/*v*) DMSO served as positive control. After stimulation, 20 µL of MTT solution (5 mg/mL in PBS, pH 7.3) was added. Then cells were incubated with MTT for 2 h at 37 °C. After removal of the supernatant, formazan crystals were dissolved by adding of 100 μL of solubilization solution (20% *w*/*v* SDS, 0.1 N HCl/DMF (1:1)). The cells with the solution were then kept in a water bath (37 °C) overnight. Then after shaking for 10 min, the absorbance was detected at a wavelength of 570 nm. The cell viability of stimulated cells was expressed as percentage cell viability in comparison to the negative control (culture medium).

## 5. Conclusions

Here we report a reliable procedure, based on complete solubilization of seed protein isolates of any origin and exhaustive enzymatic hydrolysis of glycated proteins. Thus, we overcome here the main limitation of enzymatic hydrolysis, i.e., its poor applicability to proteins, insoluble in aqueous buffers. The proposed sample preparation protocol can be efficiently coupled to the state-of-the-art LC-MS-based quantification techniques and biological assays, prospectively addressing physiological effects of glycation products, formed in seeds during ageing or/and under environmental stress. Obviously, in the future, this strategy will be easily extended to a wide panel of AGEs routinely identified by Glomb’s group (which hydrolysis procedure was used as a starting point here). Moreover, to our mind, it can be implemented in any other digestion protocols, also combined with untargeted LC-MS workflows. Finally, it might be extended to non-seed and even non-plant objects. We are convinced that in combination with rapidly developing bottom-up proteomic techniques, our method will give access to the mechanisms and pathways of the Maillard reaction in new objects and matrices, which were not available for analysis earlier.

## Figures and Tables

**Figure 1 ijms-20-03659-f001:**
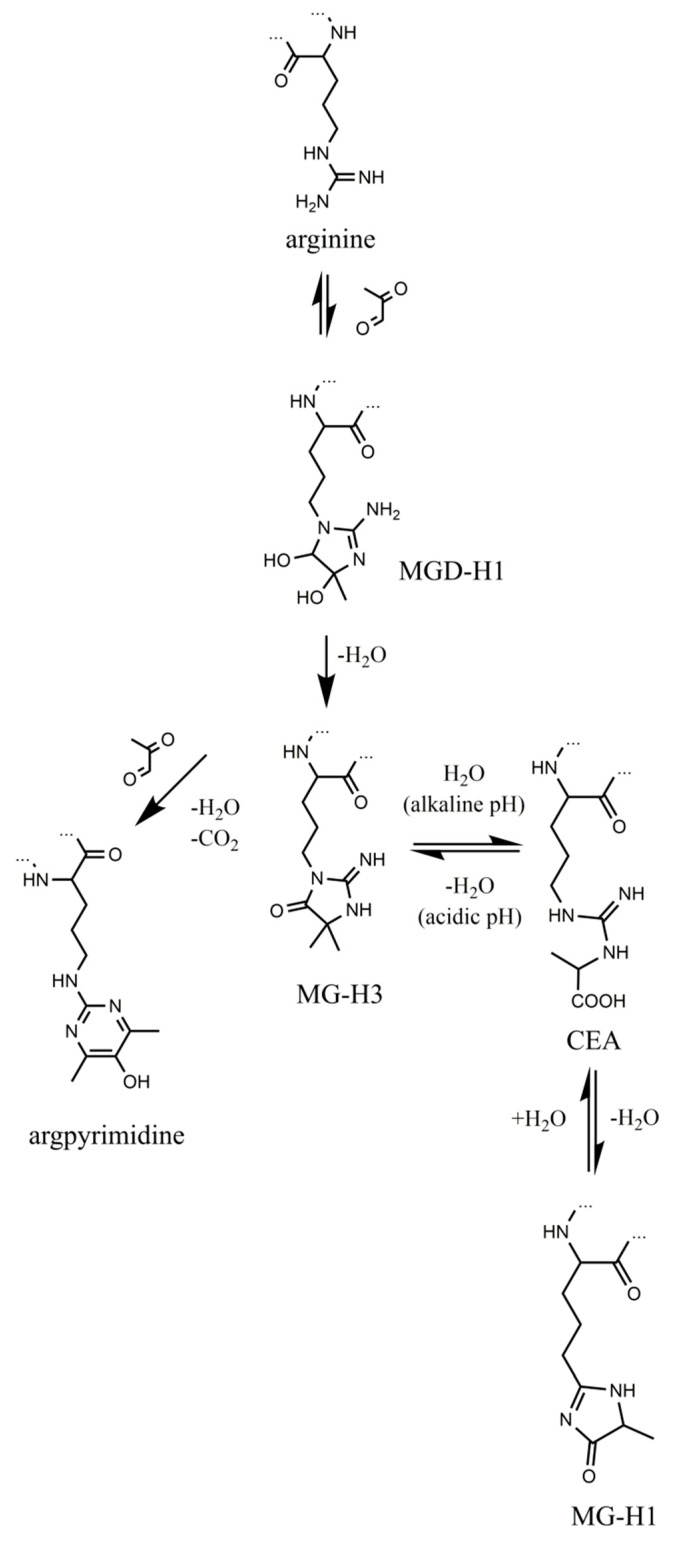
Formation and transformation of methylglyoxal-derived advanced glycation end products (AGEs). The “…” sign indicates polypeptide chain.

**Figure 2 ijms-20-03659-f002:**
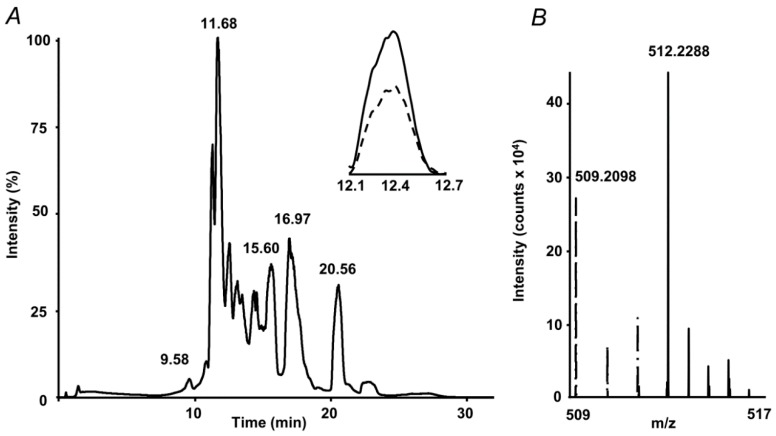
Absolute quantification of methylglyoxal hydroimidazolone 1 (MG-H1) adducts in protein hydrolysates by stable isotope dilution: total ion chromatogram (TIC) with extracted ion chromatograms (XICs) at *m*/*z* 509.21 ± 0.03 (dashed) and 512.22 ± 0.03 (solid) corresponding to the [M+H^+^]^+^ ions of MG-H1 and spiked MG-H1-d3 internal standard, respectively (**A**), and the corresponding segment of the mass spectrum, representing isotopic patterns of both isotopomers (**B**).

**Figure 3 ijms-20-03659-f003:**
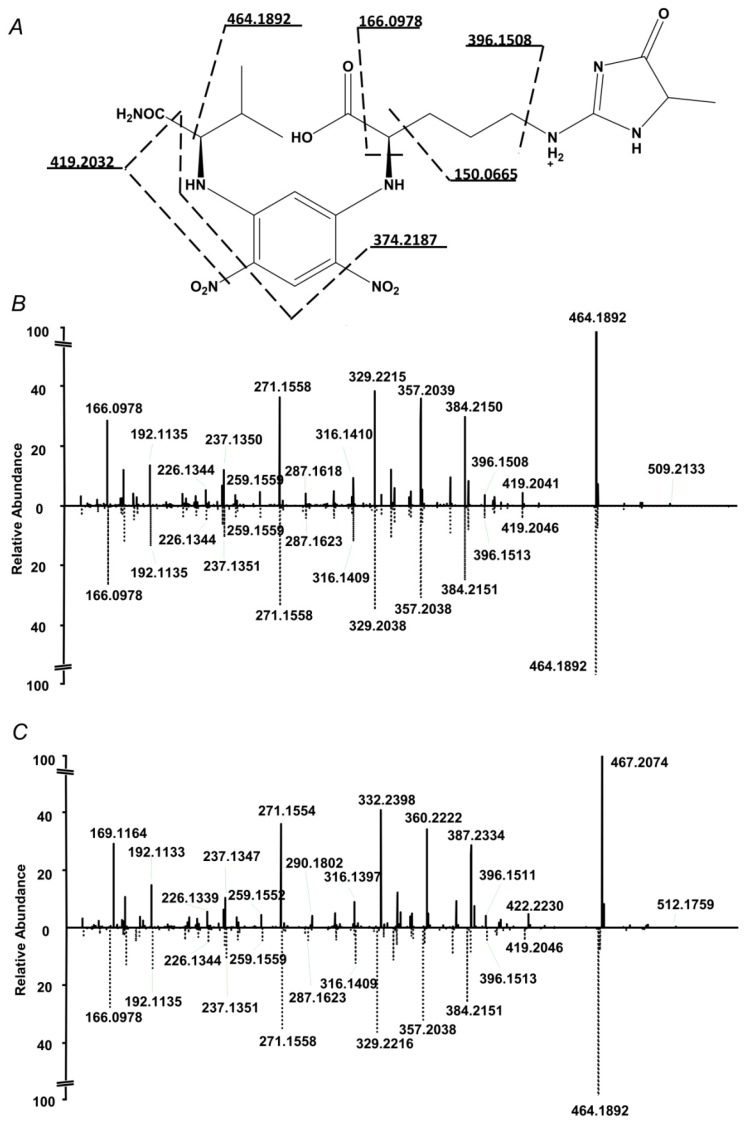
Fragmentation pattern of the *N*^2^-(5-fluoro-2,4-dinitrophenyl)-l-valine amide (L-FDVA) derivative of MG-H1 in enzymatic hydrolysates of glycated BSA (**A** and **B**,**C** bottom) and its comparison with MS/MS spectrum of authentic MG-H1 (**B** top) and internal MG-H1-d3 (**C** top) standards.

**Figure 4 ijms-20-03659-f004:**
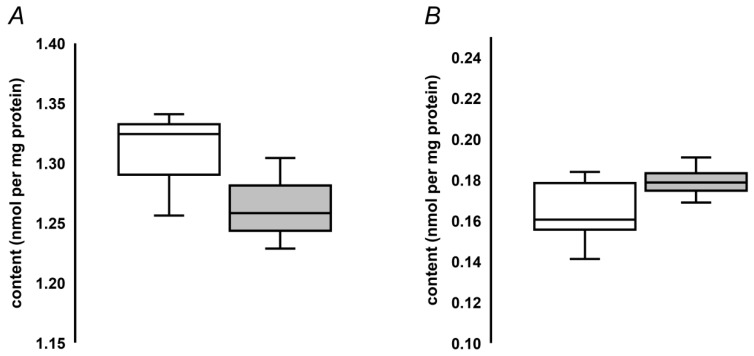
Quantification of MG-H1 adducts in pea seeds, subjected to accelerated ageing (5 days, 86% relative humidity, 45 °C, grey bars, *n* = 3) along with corresponding untreated controls (white bars, *n* = 3, **A**), and oilseed rape seeds subjected to natural ageing at 18°C during nine years (grey bars, *n* = 5) along with corresponding controls stored during one year (white bars, *n* = 5, **B**). The bars indicate median with the minimal and maximal values of the corresponding inter-quartile ranges.

**Figure 5 ijms-20-03659-f005:**
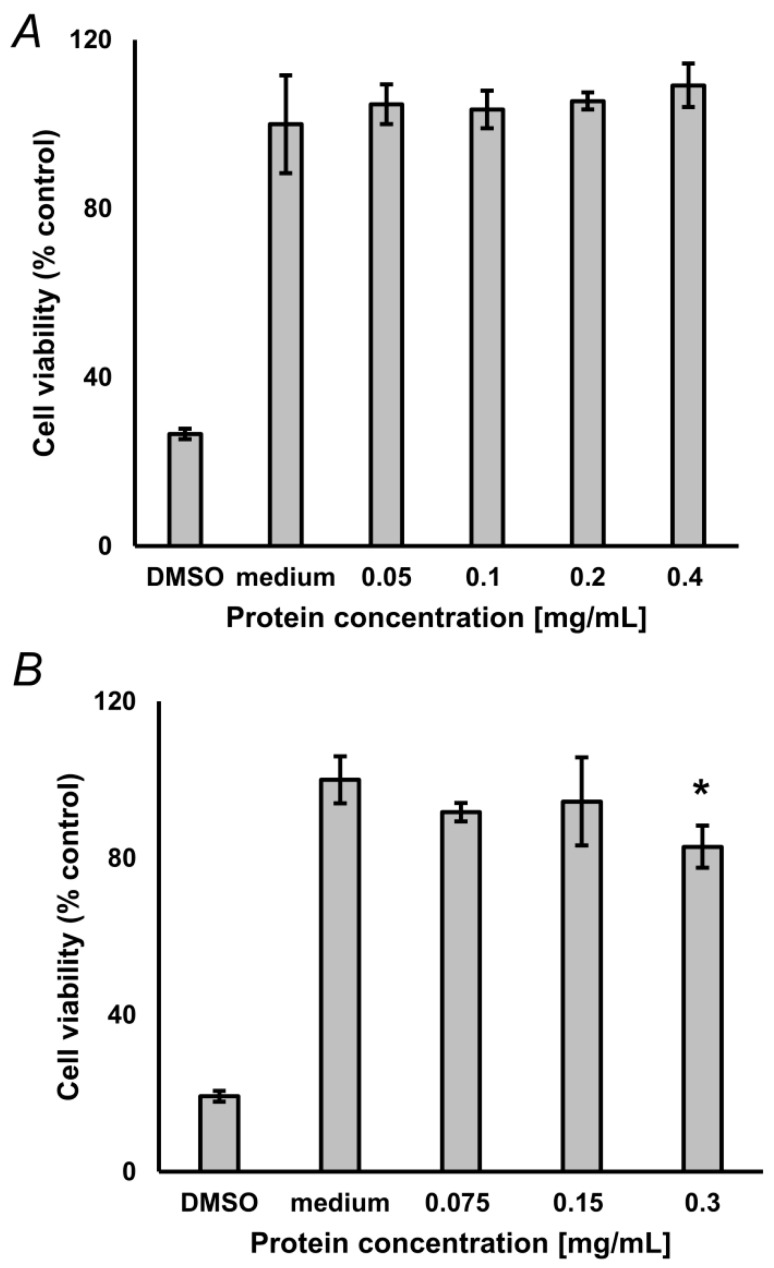
Results of the 3-(4,5-dimethylthiazol-2-yl)-2,5-diphenyltetra-zoliumbromid (MTT) assay, performed with 0.05–0.4 mg/mL glycated bovine serum albumin (BSA, *n* = 3, **A**) and 0.075–0.3 mg/mL protein isolated from pea seeds (*n* = 3, **B**), not subjected to accelerated ageing. Analyses were performed in triplicates. Culture medium served as negative control. DMSO served as positive control. * Difference was significant (*t*-test) at the confidence level of *p* < 0.05).

**Table 1 ijms-20-03659-t001:** Sensitivity and linearity parameters obtained for MG-H1 and reference amino acids.

Analyte	*m*/*z*	t_R_	LOD (fmol)	LOQ (pmol)	LDR	Slope	Intercept	R^2^
MG-H1-d3 ^a^	512.23	12.4	2.5	0.025	4.0 × 10^4^	5.0 × 10^6^	1.0 × 10^7^	0.996

^a^ Serial dilutions (*n* = 3) were prepared with the hydrolysate of 3 mg/mL glycated bovine serum albumin (BSA).

**Table 2 ijms-20-03659-t002:** Validation of the *N*^2^-(5-fluoro-2,4-dinitrophenyl)-l-valine amide (L-FDVA) derivatization of the MG-H1 adduct with intraday and inter-day precision values determined for MG-H1 in enzymatic hydrolysates of glycated BSA *^a^*.

Parameter	Intraday Precision (*n* = 4) *^b^*	Inter-Day Precision (*n* = 4/Day) *^c^*
*t*_R_ (min) ± SD (RSD%)	12.3 ± 0.024 (0.196)	12.3 ± 0.016 (0.129)
content (nmol/mg protein) AV ± SD (RSD%)	87.88 ± 0.88 (1.01)	87.35 ± 0.8 (0.92)

*^a^* Validation was performed with a hydrolysate of glycated BSA spiked with internal standard MG-H1-d_3_ (25 pmol per injection); *^b^* the maximal RSD% value, among those acquired on each of four consecutive validation days, is shown; *^c^* the inter-day precision was evaluated on four consecutive days. AV, average value; ME, mean error; RSD%, relative standard deviation percentage.

**Table 3 ijms-20-03659-t003:** Validation of RP-UHPLC-ESI-LIT-Orbitrap-MS quantification method with intraday and inter-day precision values determined for MG-H1 in enzymatic hydrolysates of glycated BSA *^a^*.

Parameter	Intraday Precision (*n* = 5) *^b^*	Inter-Day Precision (*n* = 5/Day) *^c^*
*t*_R_ (min) ± SD (RSD%)	12.3 ± 0.022 (0.179)	12.3 ± 0.014 (0.117)
content (nmol/mg protein) AV ± SD (RSD%)	86.72 ± 1.03 (1.185)	86.64 ± 0.95 (1.093)

*^a^* Validation was performed with a hydrolysate of glycated BSA spiked with internal standard MG-H1-d_3_ (25 pmol per injection); *^b^* the maximal RSD% value, among those acquired on each of four consecutive validation days, is shown; *^c^* the inter-day precision was evaluated on four consecutive days. AV, average value; ME, mean error; RSD%, relative standard deviation percentage.

**Table 4 ijms-20-03659-t004:** Validation of the sample preparation procedure (enzymatic hydrolysis with subsequent solid phase extraction) with intraday and inter-day precision values determined for MG-H1 in enzymatic hydrolysates of seed protein extract *^a^*.

Parameter	Intraday Precision (*n* = 3)	Inter-Day Precision (*n* = 3/Day) *^b^*
*t*_R_ (min) ± SD (RSD%)	12.4 ± 0.033 (0.263)	12.4 ± 0.0185 (0.15)
content (nmol mg^−1^protein) AV ± SD (RSD%)	0.73 ± 0.052 (7.66)	0.708 ± 0.037 (5.22)

*^a^* Validation was performed with a hydrolysate of seed protein extract spiked with internal standard MG-H1-d_3_ (50 pmol per injection); *^b^* the maximal RSD% value, among those acquired on each of four consecutive validation days, is shown. AV, average value; ME, mean error; RSD%, relative standard deviation percentage.

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
