# Peer review of "Analysis of Chemically Labile Glycation Adducts in Seed Proteins: Case Study of Methylglyoxal-Derived Hydroimidazolone 1 (MG-H1)"

_ijms, 2019, doi:10.3390/ijms20153659_

Reviewer 1 Report

This manuscript by Antonova et al. describes a new analysis method of protein glycation adducts. This proteinase based method is sensitive and quantitative, and can be used to analyze variety of biological samples. This study is well-done, however, I have some comments.

Chapters 2.1. and 2.2. have lots of (unnecessary) data that could be presented in supplementary data.

Figure 1: MG-H1 structure is missing one oxygen atom

Figure 1: MGD-H1 transformation to MG-H3: is one water molecule eliminated? In figure MG-H3 has one oxygen atom less that MGD-H1.

Figure 6: what grey and white bars indicate (please specify)?

Author Response

We thank the reviewer for the thoughtful review and highly appreciate the valuable comments and suggestions to improve the manuscript. Following these advices we performed all required changes in corresponding sections, as indicated in the following rebuttal addressing each aspect.

Reviewer: 1

Remarks

Remark 1: “Chapters 2.1. and 2.2. have lots of (unnecessary) data that could be presented in supplementary data.

Answer: According to the reviewer’s suggestion, we moved Figures 2 and 3 to the Supplementary information and updated the Figure numbering both in the Text and Supplementary accordingly. We found some other information in the chapters rather important for successful reproduction of our method. As we are not exceeding acceptable limits in text, we would prefer to leave these data in the main text.

Remark 2: “Figure 1: MG-H1 structure is missing one oxygen atom

Answer: We apologize for the confusing legend, provided for this figure. Actually, the structure is correct, but we did not provide sufficient information in the accompanying figure legend. Thus, we used a “…” sign to indicate that we are talking not about a free amino acid, but about a residue in a (poly)peptide chain. To avoid this confusing situation, we added this explanation now:

“The “…” sign indicates polypeptide chain” ” (lines 85 - 86).

Remark 3: “Figure 1: MGD-H1 transformation to MG-H3: is one water molecule eliminated? In figure MG-H3 has one oxygen atom less that MGD-H1

Answer: The figure is corrected accordingly.

Remark 4: “Figure 6: what grey and white bars indicate (please specify)?

Answer: We specified this in the figure legend now. Please see the corrected legend (p. 9):

Figure 6 Quantification of MG-H1 adducts in pea seeds, subjected to accelerated ageing (5 days, 86% relative humidity, 45° C, grey bars, n = 3) along with corresponding untreated controls (white bars, n = 3, A), and oilseed rape seeds subjected to natural ageing at 18° C during nine years (grey bars, n = 5) along with corresponding controls stored during one year (white bars, n = 5, B). The bars indicate median with the minimal and maximal values of the corresponding inter-quartile ranges.”

Reviewer 2 Report

The submitted study describes an improved sample preparation procedure for the mass analysis of glycation adducts in proteins, namely, enzymatic hydrolysis in the presence of detergent SDS. Since the proposed procedure could overcome the limitations of previously existing two procedures such as acid hydrolysis or enzymatic hydrolysis without detergent, the submitted study has certain usefulness to be published in International Journal of Molecular Sciences. However, some issues to be discussed are presented before the publication.  Please reconsider the manuscript based on the comments listed below.

1.       To prove usefulness of the proposed procedure in the submitted study, the detecting capability for glycation adducts using the proposed procedure should be compared with that of the previously existing procedures. Seed protein sample is prepared according to these three procedures and is loaded into same LC-MS system to identify glycation adducts as shown in Figure 1 based on their accurate mass and fragmentation pattern. If the proposed procedure has prominent detecting capability owing to mild hydrolysis conditions or good solubility of proteins, the number of identifiable glycation adducts obtained by the proposed procedure should be higher than that obtained by other procedures.

2.       The validation study of LC-MS in the submitted paper should be insufficient because the intraday and inter-day precisions were only evaluated. At least, recovery (accuracy) and matrix effect should be evaluated (K. Slimani et al., Journal of Chromatography A, 1517 2017, pages 86-96). The recovery of the proposed method should be evaluated by analyzing the protein sample spiking with the standard solution of MG-H1 in order to confirm the stability of the glycation adducts during the proposed sample preparation procedures. In addition, the matrix effect should be evaluated by comparison of the peak response of the standard solution of MG-H1 with that of the MG-H1 dissolved in the protein sample after the preparation according to the proposed procedure.

3.       The significance of the cell assays in the submitted study was unclear (Fig. 7). It seems that the results of cell assays merely explains the pea seeds protein has weak toxicity at higher concentrations. In order to confirm the compatibility of the proposed preparation procedure, the pea seeds protein isolated by previous procedures should be applied to the same cell assays and the results should be compared to the pea seeds proteins isolated by the proposed procedure.

4.       In line 342-344 of page 13, the authors described that high contents of ammonium acetate in sample can potentially affect analyte retention in ion pair chromatography. However, only loading 100 µL of sample solution into LC system (line 468 of page 16) should be insufficient amount that can influence on the retention behavior.

5.       Since the target analyte of the submitted study is glycation adducts, the sensitivity and linearity parameters can be omitted from Table 1 and related descriptions in page 8-9.

Author Response

We thank the reviewer for the thoughtful review and highly appreciate the valuable comments and suggestions to improve the manuscript. Following these advices we performed all required changes in corresponding sections, as indicated in the following rebuttal addressing each aspect.

Reviewer: 2

Remarks

Major remark 1: “To prove usefulness of the proposed procedure in the submitted study, the detecting capability for glycation adducts using the proposed procedure should be compared with that of the previously existing procedures. Seed protein sample is prepared according to these three procedures and is loaded into same LC-MS system to identify glycation adducts as shown in Figure 1 based on their accurate mass and fragmentation pattern. If the proposed procedure has prominent detecting capability owing to mild hydrolysis conditions or good solubility of proteins, the number of identifiable glycation adducts obtained by the proposed procedure should be higher than that obtained by other procedures

Answer: We agree with the reviewer, that it would be nice to give some data for a reference method, but it is not possible in our specific case – there is no reference method for analysis of hydroimidazolone AGEs in seed protein, the method reported here is the first one, proposed for seeds. We realize, that harsh conditions of acid hydrolysis destroy hydroimidazolones. This fact is well-proved (Ahmed et al. Biochem J 2002, 364, 1–14.), and now is totally accepted in the IMRS (International Maillard Reaction Society) – acidic hydrolysis should not be considered when contents of hydroimidazolones are addressed. On the other hand, to our mind, hydrolysis and following LC-MS/MS analysis is totally senseless in the case of incomplete dissolution of protein – in this case we will not know, what proteins are hydrolyzed, and what not, and biological interpretation of results will be speculative. Based on these considerations, we decided not to analyze MG-H1 with the techniques, already known to be not appropriate for this.

Major remark 2: “The validation study of LC-MS in the submitted paper should be insufficient because the intraday and inter-day precisions were only evaluated. At least, recovery (accuracy) and matrix effect should be evaluated (K. Slimani et al., Journal of Chromatography A, 1517 2017, pages 86-96). The recovery of the proposed method should be evaluated by analyzing the protein sample spiking with the standard solution of MG-H1 in order to confirm the stability of the glycation adducts during the proposed sample preparation procedures. In addition, the matrix effect should be evaluated by comparison of the peak response of the standard solution of MG-H1 with that of the MG-H1 dissolved in the protein sample after the preparation according to the proposed procedure.

Answer: We agree with the reviewer – these data would improve this manuscript. Therefore, we provide recovery and matrix effect data as required. For assessment of matrix effect we used a procedure of Böttcher et al. 2007, Anal. Chem. 79, 1507-1513 conventionally used for this purpose in our institute. To address all issues, pointed out by the reviewer, we added Figures S-9, S-10 and S-11 to the Supplementary information and extended the main text with the following paragraphs:

“For analysis of recovery, MG-H1-d3 was spiked to isolated pea protein in aq. 10 % (w/v) SDS prior to dilution with PBS and enzymatic hydrolysis. To address the interference of the target analyte with the components of protein hydrolyzates, the recovered amounts of MG-H1-d3 were compared to those, observed when the same aliquots of MG-H1-d3 were spiked to aq. 10 % (w/v) SDS, subjected to the identical treatment. To address the effect of incubations with the enzymes, the recovered amounts of MG-H1-d3 were compared to the standard solution of MG-H1-d3, dissolved in PBS at the same concentration. The analysis revealed relatively low contribution of other hydrolysis products (i.e. highly abundant proteinogenic amino acids): a slight enhancement of detection was observed in comparison to blank hydrolysis samples (recovery 109.4%, Figure S-9). In contrast, the prolonged hydrolysis procedure itself resulted in a relatively high degree of the target analyte suppression or degradation – only 18% of MG-H1-d3 could be recovered, although this result was reliably reproducible (Figure S-9).

For analysis the of possible matrix effects as a potential reason for the observed recovery loss, the approach of Böttcher et al. was applied [ ]. For this, the MG-H1-d3 standard (0.2 µmol/L) was infused with a syringe pump at 3 µL/min in the effluent of RP-HPLC column after injection of blank (20% v/v aq. acetonitrile in 0.1% v/v formic acid, Figure S-10) or derivatized pea hydrolyzate (Figure S-11). Although no matrix effects could be detected in blank (Figure S-10B), approximately complete suppression of the MG-H1-d3 signal was observed at tR 8.8 – 9.2 min and 11.5 – 12.0 min when the standard was infused in the column effluent obtained with pea protein hydrolyzate (Figure S-11B). Thus, the section of the chromatogram, corresponding to tR of MG-H1 and its labeled counterpart (tR = 12.3 min) is only minimally affected by ion suppression, i.e. the observed loss of recovery upon the hydrolysis procedure is mostly related to the poor stability of the hydroimidazolone glycation adduct under the prolonged hydrolysis conditions” (Lines 232 - 254).        

Major remark 3: “The significance of the cell assays in the submitted study was unclear (Fig. 7). It seems that the results of cell assays merely explains the pea seeds protein has weak toxicity at higher concentrations. In order to confirm the compatibility of the proposed preparation procedure, the pea seeds protein isolated by previous procedures should be applied to the same cell assays and the results should be compared to the pea seeds proteins isolated by the proposed procedure.

Answer: We agree with the reviewer – obviously, our rationale for cell assays remained unclear for the reader, and the message needs to be modified appropriately. Thus, the aim of cell assays was to show, that our SDS removal procedure can give access to cell assays. For this, it was necessary to clear, if the hydrolyzates after the SPE procedure are toxic or not anymore. Before the SPE procedure, the concentration the concentration of SDS was 0.05 % (w/v) that is close to 1%, usually used for protein isolation from cells, i.e. these conditions are incompatible. We extended the text, to make our considerations more clear:

“The main aim of this section was to verify, that the degree of SDS removal with our SPE procedure is sufficient for cell assays, i.e. that residual SDS traces do not exert cytotoxicity, or such effects are negligible”  (Lines 264 - 266).       

It needs to be once more noted: the proposed method is the first one, designed for analysis of hydroimidazolone modifications upon exhaustive enzymatic protein hydrolysis. No such methods were reported for seed proteins before. Therefore, no methods for comparison as a reference can be chosen, to our mind.

Major remark 4: “In line 342-344 of page 13, the authors described that high contents of ammonium acetate in sample can potentially affect analyte retention in ion pair chromatography. However, only loading 100 µL of sample solution into LC system (line 468 of page 16) should be insufficient amount that can influence on the retention behavior

Answer: We agree with the reviewer, is not properly explained. Indeed, not only injection volume, but also flow rate is important. We apologize about it and improve our text accordingly:

“Indeed, due to a relatively low flow rate, a local drop of the ion pair reagent concentration in the eluent can occur, and retention time shifts can be observed (personal unpublished observation). Thereby, higher injection volumes (as, for example, 100 µL used here) would favor these effects.” (Lines 378 - 381).

And further:

“As reduction of injection volumes seems to be not possible in terms of sensitivity, an additional SPE step might be considered.” ( Lines 383 - 384).

Major remark 5: “Since the target analyte of the submitted study is glycation adducts, the sensitivity and linearity parameters can be omitted from Table 1 and related descriptions in page 8-9

Answer: Corrected accordingly, the values for reference amino acids are moved to supplementary (Line 198). Please see also Table S-3 added to the Supplementary.

Round  2

Reviewer 2 Report

Although the authors revised the manuscript sufficiently according to the comments, there was a single issue to be discussed. To obtain the results shown in Fig. 1, intact (non-derivatized) MG-H1-d3 was infused, not L-FDVA derivative of MG-H1-d3. Is this manner correct?

Author Response

We thank the reviewer for noting this bug: indeed, it would be wrong to use non-derivatized glycation adduct. Of course, we derivatized MG-H1-d3 with L-FDVA and infused the derivative, filled in the syringe at the given concentration. We apologize because this mistake, related to writing, but not experimental setup. We made a correction in text:

"For analysis the of possible matrix effects as a potential reason for the observed recovery loss, the approach of Böttcher et al. [38] was applied. For this, a L-FDVA derivative of the MG-H1-d3 standard (0.2 µmol/L) was infused with a syringe pump at 3 µL/min in the effluent of RP-HPLC column after injection of blank (20% v/v aq. acetonitrile in 0.1% v/v formic acid, Figure S-10) or derivatized pea protein hydrolyzate (13.6 µg, Figure S-11). Although no matrix effects could be detected in blank (Figure S-10B), approximately complete suppression of the MG-H1-d3-L-FDVA signal was observed at tR 8.8 – 9.2 min and 11.5 – 12.0 min when the standard was infused in the column effluent obtained with pea protein hydrolyzate (Figure S-11B). " (lines 243 - 247).